# Nanosecond Pulsed Electric Fields Induce Endoplasmic Reticulum Stress Accompanied by Immunogenic Cell Death in Murine Models of Lymphoma and Colorectal Cancer

**DOI:** 10.3390/cancers11122034

**Published:** 2019-12-17

**Authors:** Alessandra Rossi, Olga N. Pakhomova, Peter A. Mollica, Maura Casciola, Uma Mangalanathan, Andrei G. Pakhomov, Claudia Muratori

**Affiliations:** 1Frank Reidy Research Center for Bioelectrics, Old Dominion University, Norfolk, VA 23508, USA; rossi.andra@gmail.com (A.R.); opakhomo@odu.edu (O.N.P.); mcasciol@odu.edu (M.C.); umangala@odu.edu (U.M.); apakhomo@odu.edu (A.G.P.); 2Department of Medical Diagnostics and Translational Sciences, Old Dominion University, Norfolk, VA 23508, USA; pmollica@odu.edu

**Keywords:** ablation, immunotherapy, vaccine, apoptosis, electroporation

## Abstract

Depending on the initiating stimulus, cancer cell death can be immunogenic or non-immunogenic. Inducers of immunogenic cell death (ICD) rely on endoplasmic reticulum (ER) stress for the trafficking of danger signals such as calreticulin (CRT) and ATP. We found that nanosecond pulsed electric fields (nsPEF), an emerging new modality for tumor ablation, cause the activation of the ER-resident stress sensor PERK in both CT-26 colon carcinoma and EL-4 lymphoma cells. PERK activation correlates with sustained CRT exposure on the cell plasma membrane and apoptosis induction in both nsPEF-treated cell lines. Our results show that, in CT-26 cells, the activity of caspase-3/7 was increased fourteen-fold as compared with four-fold in EL-4 cells. Moreover, while nsPEF treatments induced the release of the ICD hallmark HMGB1 in both cell lines, extracellular ATP was detected only in CT-26. Finally, in vaccination assays, CT-26 cells treated with nsPEF or doxorubicin equally impaired the growth of tumors at challenge sites eliciting a protective anticancer immune response in 78% and 80% of the animals, respectively. As compared to CT-26, both nsPEF- and mitoxantrone-treated EL-4 cells had a less pronounced effect and protected 50% and 20% of the animals, respectively. These results support our conclusion that nsPEF induce ER stress, accompanied by *bona fide* ICD.

## 1. Introduction

Several local ablative therapies are available as innovative treatments to fight cancer. Although both ablation and surgical resection result in the focal eradication of tumors, ablation causes cell death in situ. By killing tumors locally, ablative therapies make tumor antigens available as an in situ cancer vaccine which has the potential to initiate a systemic antitumor immune response. 

Thermal ablation technologies such as radiofrequency ablation (RFA), microwave ablation (MWA), and high intensity focused ultrasound (HIFU) have all been shown to elicit immunomodulatory effects, the magnitude of which is highly variable [1]. While each modality accomplishes the local deposition of energy via different approaches, all these ablation methods cause cell death via necrosis. Accidental necrosis is a rapid cell death process which lacks morphological and biochemical changes that are typically associated with late apoptosis and secondary necrosis [2]. Although in vitro damage associated molecular patterns (DAMPs) released from necrotic cells led to maturation of antigen presenting cells (APCs), in vivo necrotic cell vaccines showed an immunologically inert nature which did not protect animals from tumor challenge [3,4]. 

The potential of dead and dying cells to trigger an antitumor immune response depends on the regulated emission of DAMPs, such as the endoplasmic reticulum protein calreticulin (CRT), ATP, and the chromatin-binding protein high mobility group B1 (HMGB1). These signals are crucial for immunogenic cell death (ICD). The extracellular release of HMGB1 and ATP attracts and activates APCs, whereas CRT exposure on the surface of dying tumor cells serves as an “eat-me” signal for phagocytes [5,6,7]. Collectively, these signals enhance APCs ability to engulf, process, and present tumor-derived antigens to T-cells, thus favoring the induction of a tumor-specific adaptive immunity [5,8,9]. 

Several studies have shown that the stress responses that precede cell death control the temporally coordinated emission of DAMPs [5,6,10]. Among these stress responses, the central role for endoplasmic reticulum (ER) stress has been revealed in all scenarios of ICD described thus far [11]. The endoplasmic reticulum is a crucial organelle that controls Ca^2+^-storage and buffering, lipid biosynthesis, and folding and assembling of secretory and transmembrane proteins. Considering the relevance of maintaining ER homeostasis, cells have developed an adaptation mechanism to rapidly sense and respond to perturbations in the ER folding machinery, called the unfolded protein response (UPR) [11,12,13]. In mammalian cells, the UPR is controlled by three ER-resident stress sensors, namely, inositol-requiring enzyme 1 alpha (IRE1α), activating transcription factor 6 (ATF6), and protein kinase R-like ER kinase (PERK). When the ER stress is mild, the UPR facilitates the re-establishment of the cellular homeostasis. However, when ER stress intensity is too severe, the UPR engages signaling pathways culminating into cell death executed through the apoptotic program [11,12,14,15]. 

Considering the dependence of ICD on the induction of sustained ER stress, it is no surprise that the ER stress intensity caused by an ICD inducer has a strong impact on the quality and quantity of the emitted DAMPs. This led to the subdivision of ICD inducers into two categories, Type I and type II. Type I ICD inducers, such as anthracyclines, oxaliplatin, and radiotherapy, trigger apoptotic cell death which is accompanied by collateral ER-stress. Type II inducers, such as hypericin-photodynamic therapy (Hyp-PDT) or oncolytic viruses, selectively targets the ER, inducing ER stress-dependent apoptosis [10,16,17,18]. Among the three ER-stress sensors, PERK has been shown to have a predominant role in mediating the trafficking of DAMPs. PERK regulates the externalization of CRT in response to chemotherapy [7,19,20,21] and the emission of both CRT and ATP after Hyp-PDT [22]. Moreover, ablation of PERK in cancer cells compromises ICD by suppressing its tumor-rejecting anticancer vaccination effect in vivo [19]. 

Pulsed electric fields (PEF) compromise the integrity of the cell plasma membrane, a process referred to as electroporation or electropermeabilization [23,24,25]. Conventional electroporation pulses are relatively long (µs to ms) with lower electric fields (≤1  kV/cm) and they have been widely used in many biomedical applications including in vitro cell transfection [26], in vivo gene transfer [27,28], and intracellular delivery of cytotoxic agents such as bleomycin to kill tumors [29,30,31,32]. Ablation of tumors can also be achieved by more intense PEF treatments which cause irreversible electroporation (IRE) [33,34,35]. Treatment of tissue with PEF requires the application of electrodes to the area to be treated and the delivery of PEF according to specific protocols. The treatment intensity (as determined by the pulse amplitude, number, and duration) needs to be sufficiently high to destroy the cells, but not too high in order to avoid the concurrent Joule heating and thermal damage.

Recently, PEF protocols have been extended to nanosecond-duration pulses (nsPEF). Unlike micro- and millisecond duration PEF, under specific conditions nsPEF have been shown to penetrate the cell plasma membrane and permeabilize organelle membranes such as the ER, mitochondria, and nucleus [36,37,38,39,40,41]. In the conventional electroporation, charging of the cell plasma membrane compensates the external electric field and protects the cell interior. Characteristic charging time constants for the plasma membrane of mammalian cells are in the order of 100 ns. However, for field magnitudes greater than 10 kV/cm, ultrashort nanosecond pulses can charge smaller intracellular structures to the electroporation threshold faster than it would take for the plasma membrane to charge and protect them [42,43,44]. Subsequent biological responses range from Ca^2+^ mobilization [39,40,41,45,46], reactive oxygen species (ROS) production [47,48], rapid externalization of phosphatidylserine [49,50,51], activation of the c-Jun N-terminal kinase 1 (JNK) [52], and mitogen-activated protein kinase (MAPK) [53] pathways to the induction of apoptotic and necrotic cell death [37,54,55,56,57].

Cancer cell killing by nsPEF has been extensively explored in vitro [37,55,57,58,59,60] followed by successful tumor ablation trials in vivo in animals [57,61,62,63,64,65,66] and humans [67]. 

In addition to the high ablation efficiency, several groups, including ours, have reported that nsPEF elicit an antitumor immune response which, depending on the tumor model or pulse parameters, protected 100% to 25% of the treated animals from tumor challenge [57,65,66,68,69,70,71]. The mechanisms responsible for this inhibition remain poorly understood. 

In this study, we investigated the efficacy of nsPEF treatment at inducing ER stress and ICD in both immunogenic CT-26 colon carcinoma and poorly immunogenic EL-4 lymphoma cells. Our results show that nsPEF activate PERK, accompanied by apoptotic cell death and emission of major DAMPs. Moreover, nsPEF-treated CT26 and EL-4 tumor cells vaccinated 78% and 50% of the animals from tumor challenge, respectively. 

## 2. Results

### 2.1. nsPEF Induce ER-Stress in Cancer Cells

Because nsPEF have been shown to directly permeabilize the ER leading to a rise in cytosolic Ca^2+^ [39,40] and to induce ROS production [47,48], we tested a hypothesis that nsPEF treatments could trigger a generalized ER stress response in tumor cells. 

In preliminary experiments we investigated the sensitivity to nsPEF of both immunogenic CT-26 and poorly immunogenic EL-4 cells and established 300 and 100 pulses (200 ns, 7 kV/cm, 10 Hz) as iso-effective doses causing 60% cell death at 24 h, respectively. Using these pulse doses, we measured the level of spliced *XBP1* mRNA in both nsPEF-treated tumor cell lines. XBP1 is a key transcription factor that regulates the UPR. Its expression is regulated by unconventional mRNA splicing that is carried out by the ER-sensor IRE1 [72,73]. Figure 1A shows that in EL-4 cells (top panel) 200 ns pulses did not induce an accumulation of spliced *XBP1*. Conversely, in CT-26 cells (bottom panel) nsPEF increased the level of spliced *XBP1* by five-fold.

During ICD, while all three sensors of the UPR are activated, only the PERK branch is so far mandatory for ICD [19,21]. Therefore, we studied PERK activation in nanoporated cells. Our results show that 200 ns pulses induce the activation of PERK in both cell lines as measured by the phosphorylation of its substrate eukaryotic translation initiation factor 2α (eIF2α) on serine 51 (Figure 1B). 

Overall our data show that nsPEF induce ER stress accompanied by PERK activation while the IRE1 branch of the UPR response appear to be activated in a cell type-specific manner.

### 2.2. PERK Activation Correlates with Apoptosis Induction and CRT Translocation onto the Cell Plasma Membrane of Nanoporated Tumor Cells

Unresolved ER stress triggers apoptotic cell death. Phosphorylation of eIF2α by PERK induces the proapoptotic transcription factor CAAT/enhancer binding protein (C/EBP) homologous protein (CHOP), which is a pivotal element of the switch from pro-survival to pro-death signaling [14,15,74]. We previously reported that nsPEF induce apoptosis in U-937 monocyte lymphoma but not in B16F10 melanoma cells [57]. We, therefore, decided to investigate apoptosis in both CT26 and EL-4 cells and compare it to B16F10 melanoma cells.

Cells were exposed to either 200 ns pulses (Figure 2A,B) or staurosporin as a positive control for apoptosis induction (Figure 2C), and caspase-3/7 activity and viability were measured at 4 and 24 h post treatment. The use of different numbers of pulses for the three cell lines reflects their different sensitivity to nsPEF, with EL-4 and B16F10 being more sensitive than CT26 cells. Figure 2A shows that nsPEF caused caspase activation at 24 h in both EL-4 and CT26 while, as expected, failed to do so in B16F10 cells. In CT-26 cells the activity of caspase-3/7 increased fourteen-fold as compared to four-fold in EL-4 cells. Notably, cell death kinetics vary considerably between cell lines (Figure 2B). While both EL-4 and CT26 cells kept dying over 24 h, suggesting the occurrence of a slower programmed cell death, B16F10 cells died rapidly within the first 4 h, and in 24 h, cells already started to grow back (Figure 2B). Notably, primary necrosis can contribute to nsPEF cytotoxicity [54] explaining why overall cell death was similar in all treated cell types. 

Previous studies have shown that PERK activation and the phosphorylation of its substrate eIF2α are essential for CRT exposure induced by anthracyclines, oxaliplatin, and Hyp-PDT [19,22]. Therefore, we investigated whether PERK activation correlated with CRT exposure on the plasma membrane of nanoporated tumor cells. Figure 3 shows that nsPEF triggered CRT translocation in both EL-4 (Figure 3A) and CT26 (Figure 3B). Notably, the efficiency of nsPEF at inducing CRT relocalization to the cell plasma membrane was comparable or even higher than the effect of the positive control mitoxantrone. 

### 2.3. Effect of nsPEF on Secreted Immunogenic DAMPs

Next, we analyzed the effect of nsPEF on DAMPs which are released in the cell supernatant during ICD such as the nuclear protein HMGB1 and ATP (Figure 4). Our results show that while nsPEF treatments induced HMGB1 release in both tumor cell lines (Figure 4A), extracellular ATP was detected only in CT26-treated cells (Figure 4B). In CT26 cells, HMGB1 and ATP levels were higher than doxorubicin positive controls (Figure 4A,B, right graphs). Notably, ATP release was detected as late as 18 h after nsPEF exposure, suggesting that nanoporation can trigger active, persistent mechanisms for ATP secretion. 

### 2.4. Immunogenicity of nsPEF-Induced Cell Death

Our in vitro results show that nsPEF induce ER stress accompanied by apoptosis and emission of major DAMPs. The capacity of nsPEF to induce *bona fide* ICD was finally tested in standard vaccination experiments. CT26 and EL-4 cells were treated with 600 and 200 pulses (200 ns, 7 kV/cm, 10 Hz), respectively, and in order to allow ICD to occur in vivo, immediately injected in syngeneic mice. Figure 2B shows that for both cell lines, even at the highest pulse doses, cell death leveled off to 80% to 85%. These results are consistent with previous studies showing that exposures of suspension cells in electroporation cuvettes do not result in 100% cell killing [55,58,60]. Although treated with a vaccine containing 15% to 20% live cells, tumors at vaccination sites did not develop in 60% (nine out of fifteen) and 25% (six out twenty-five) of CT-26 and EL-4 syngeneic mice, respectively. The difference between the two models may reflect their intrinsic immunogenicity with CT-26 being more immunogenic than EL-4 cells [75,76]. In animals that did not develop tumors at the vaccination site, CT26 cells treated with nsPEF and doxorubicin equally impaired the growth of tumors at challenge sites (Figure 5A) eliciting a protective anticancer immune response in 78% (seven out of nine) and 80% (eight out of ten) of the animals, respectively (Figure 5B). Among animals with tumors at the primary injection site, five out of six developed tumors also at challenge sites, yet these tumors grew significantly slower (Appendix A). Compared to CT-26, nsPEF-treated EL-4 cells had a less pronounced effect and protected 50% (three out of six) of the animals (Figure 6A). Notably, both 0.5 and 1 µM mitoxantrone-treated cells failed to induce an effective antitumor immune response in EL-4 syngeneic mice (Figure 6B). Results for animals that developed tumors at vaccination site are not presented because the fast tumor growth kinetic did not allow us to monitor the animals long term after challenge. 

## 3. Discussion

In this study we show that nsPEF stimulate an anticancer immune response via induction of ICD. Although Nuccitelli et al. [71] have previously reported the effect of nsPEF on DAMPs emission, this is the first comprehensive study investigating ICD induction in nanoporated tumor cells. According to Kepp et al. [77], we measured apoptosis, ER stress, CRT exposure, ATP secretion, and HMGB1 release and proved the ICD-inducing capacity of nsPEF in standard vaccination assays in immunocompetent and syngeneic mouse models. 

Our results show that nsPEF can induce the phosphorylation of eIF2α in both CT26 and EL-4 tumor cells. The eIF2α phosphorylation is the hallmark of the integrated stress response (IRS), a complex signaling pathway that is activated in response to both cell extrinsic factors such as hypoxia, amino acid and glucose deprivation, viral infections, and cell intrinsic stresses such as ER stress [78]. Four distinct specific kinases, namely, PERK, double-stranded RNA-dependent protein kinase (PKR), heme-regulated eIF2α kinase (HRI), and general control non-derepressible 2 kinase (GCN2) can catalyze eIF2α phosphorylation [78]. However, in the context of ICD, it is the ER stress sensor PERK that phosphorylates eIF2α [19,21,22]. Moreover, Morotomi-Yano et al. have previously reported that, in HeLa cells, both PERK and GCN2 contribute to nsPEF-induced eIF2α phosphorylation causing a block in protein translation [79]. 

Our results also show that, in CT26 cells, nsPEF trigger a more generalized ER stress response by activating both IRE1 and PERK, whereas, in EL-4 cells, nsPEF fail to activate IRE1. Notably, as reported by Morotomi-Yano et al., IRE1 was also not activated in nanoporated HeLa cells. Moreover, the same study reported that nsPEF-induced eIF2α phosphorylation did not stimulate the expression of its main downstream transcription factors, namely, activating transcription factor 4 (ATF4) and CHOP [79]. Although surprisingly, these results are in agreement with the recent study from Bezu et al. who reported that the ICD inducers mitoxantrone, doxorubicin, and oxaliplatin cause PERK-mediated eIF2α phosphorylation but are unable to induce IRE-mediated splicing of *XBP1* and fail to stimulate ATF4 and CHOP expression [21]. These data suggest that an incomplete ER stress response is enough to induce ICD, although, in certain tumor cell lines, nsPEF can engage more arms of this response. 

Considering these results, a key question to consider is how nsPEF trigger ER stress. Perturbations in cellular energy, Ca^2+^ homeostasis or redox status can all cause ER stress and PERK activation. Because nanosecond pulses are distinguished by their ability to permeabilize the ER and recruit Ca^2+^ from intracellular stores [39,40,41,45,46], one can speculate that nsPEF can cause ER Ca^2+^ depletion. Notably, thapsigargin, a pharmacological antagonist of sarcoplasmic/endoplasmic reticulum Ca^2+^ ATPase (SERCA), the pump that moves Ca^2+^ into the ER, causes ER stress by draining the Ca^2+^ ER storage. This depletion decreases the activity of Ca^2+^-dependent chaperones leading to an increase in unfolded proteins and activation of the UPR signaling [12].

Most ICD stimuli including anthracyclines, UVC radiation, and Hyp-PDT trigger ROS-dependent ER stress. For instance, in Hyp-PDT, hypericin accumulates preferentially within the membranes of the ER [80] and its photoactivation generates ROS [81]. These short-lived species act locally on ER membrane phospholipids [82]. Lipid peroxidation can lead to inactivation of receptors and proteins such as SERCA during Hyp-PDT [83,84,85]. Because PEF, including nsPEF, have been found to increase the level of ROS in the cell [47,86,87], one could hypothesize that electroporation can induce ER stress through mechanisms similar to Hyp-PDT. nsPEF exposure was shown to damage mitochondria causing leakage of ROS and cytochrome C [37]. Moreover, electropermeabilization is accompanied by lipid peroxidation, as previously shown in both mammalian cells and liposomes [88,89,90,91,92]. 

In our study we show that the capacity of nsPEF to engage more arms of the ER stress response can affect the quality and quantity of emitted DAMPs. Indeed, while nsPEF trigger CRT externalization in both CT26 and EL-4, ATP was detectable only in the supernatant of nanoporated CT26 cells. Previous studies have reported that electroporation causes leakage of ATP through the pores [93]. However, in CT26 ATP release was detected 18 h after nsPEF exposure, suggesting the engagement of active mechanisms for ATP secretion in nanoporated tumor cells. Active ATP release in response to ICD inducers was shown to be dependent on autophagy [6], a catabolic process evolved to maintain the homeostasis of proteins and damaged organelles. Recently autophagy has emerged as an essential protective mechanism during ER stress and IRE1-mediated JNK phosphorylation appears to be a major regulator in this pathway. Specifically, IRE1 activation leads to JNK phosphorylation, which in turn regulates the autophagy related protein Beclin-1 (BECN1). The activation of BECN1 is followed by direct phosphorylation of B-cell lymphoma 2 (BCL2), which in turn disrupts the interaction between BECN1 and BCL2 and induces autophagy in tumor cells [94]. In addition to the kinase activity, the endoribonuclease activity of IRE1 also participates in the induction of autophagy. Spliced XBP1 binds directly to the *BECN1* promoter supporting an autophagic response via transcriptional upregulation of *BECN1* [95]. Altogether, these evidences suggest that the capacity to activate IRE1 can affect autophagy-dependent ATP secretion in nanoporated tumor cells. Indeed, although autophagy was not investigated, our results show that ATP secretion correlates with IRE1 activation by nsPEF in CT26 cells. Moreover, nsPEF can induce autophagy, as shown by Ullery et al., in both human lymphoma U937 and Chinese Hamster Ovarian-K1 (CHO-K1) cells [96]. 

The discrepancy in ATP detection in the supernatant of nanoporated CT26 and EL-4 cells can also reflect a different expression level of ATP-degrading ectoenzymes. Tumor cells are proficient at converting ATP into adenosine through the expression of ectonucleotidases such as CD39 and CD73 [97]. To support this hypothesis, our results show that even in response to positive controls such as doxorubicin, the level of ATP secretion was markedly higher in CT26 than in EL-4 cells. 

In addition to ER stress, apoptotic cell death was shown to be crucial for ICD [98]. Our study shows that nsPEF triggered apoptotic cell death in CT26 and EL-4, although to a different extent, in CT-26 cells the activity of caspase-3/7 increased fourteen-fold as compared with four-fold in EL-4 cells. Several studies have shown that nsPEF can trigger intrinsic, mitochondria-mediated, apoptotic cell death [99,100]. Notably, in the case of unresolved ER stress, both PERK and IRE1 arms of the UPR can engage the mitochondrial apoptotic machinery. Regarding PERK, phosphorylation of eIF2α favors the translation of ATF4 which in turn induces the expression of the proapoptotic factor CHOP [101]. Concerning IRE1, prolonged activation of JNK induces apoptosis by dissociating BCL2 Associated X (BAX) from BCL2 leading to BAX activation [102]. Although the mechanisms through which the UPR integrates pro-death and pro-survival signals are poorly understood, one can speculate that the capacity of nsPEF to engage more arms of the ER stress response may determinate the extent of apoptosis induction in nanoporated tumor cells. 

From the above discussion it appears that different tumor cell types can respond differently to nsPEF. This could be related to the extent of ROS formation or permeabilization of the ER achieved by the pulse treatment in a specific tumor cell type. Indeed, CT26 are more resistant to nsPEF than EL-4, suggesting that the higher doses needed to kill colon carcinoma cells could have triggered stress responses and cell death pathways which are not activated at lower pulse doses. In addition to the intensity of the pulse treatment, genetic or epigenetic alterations that compromise stress responses or cell death signaling pathways can also affect ICD induction by nsPEF. 

A challenge in our vaccination experiments was the inability to achieve complete cell killing in vitro. This failure to reach 100% cell death was previously explained by possible nonuniform electric field exposure of cells at the edges of the cuvette and in the suspension meniscus, or by possible shielding of some cells by other cells, as well as by an unusual level of resistivity to nsPEF in a limited subpopulation of cells [55,58,60]. In our experimental conditions, tumor cell vaccines contained 0.09 to 0.12 × 10^6^ (15% to 20%) live cells. Although the residual live cells in the vaccine were more than enough to establish a tumor (more than the cells used to challenge the animals), tumors at the vaccination site did not develop in 60% and 25% of CT26 and EL-4 syngeneic mice, respectively. These results challenge the recent study from Sulciner et al. where in multiple tumor models, including EL-4 cells, tumor cells killed by chemotherapy (cisplatin, vincristine, gemcitabine, or docetaxel) or targeted therapy (erlotinib or cetuximab) and co-injected with a subthreshold inoculum of live cells dramatically stimulated primary tumor growth [103]. Remarkably, none of the drugs used in the study were ICD inducers which, together with our results, suggests that the occurrence of ICD can overwhelm and counteract the cancer promoting effects of dying tumor cells. Considering the above discussion, what we initially believed to be a limitation for the vaccination study turned out to be an additional evidence that nsPEF-induced cell death is perceived by the immune system as immunogenic.

## 4. Materials and Methods

### 4.1. Cell Culture and Reagents

Both mouse EL-4 lymphoma and B16F10 melanoma cells were acquired from ATCC (ATCC, Manassas, VA, USA) and cultured in Dulbecco’s modified Eagle medium (DMEM, Corning, Corning, NY, USA). CT26 colon carcinoma cells (ATCC) were cultured in RPMI 1640 (Corning). Culture media were supplemented with L-glutamine (ATCC), 10% fetal bovine serum (Atlanta Biologicals, Norcross, GA, USA), 100 U/mL penicillin and 0.1 mg/mL streptomycin (Gibco, Gaithersburgh, MD, USA).

Thaspigargin (1 µM), mitoxantrone (0.5–1 µM), doxorubicin (10–25 µM), and staurosporin (10 µM) were all from Sigma (Sigma-Aldrich, St. Louis, MO, USA). 

### 4.2. Pulsed Electric Field Exposure Methods

Cell samples were resuspended at 1.2 to 3 × 10^6^ cell/mL in the growth medium or PBS and loaded in 1 mm gap electroporation cuvettes (BioSmith, San Diego, CA, USA). Samples were subjected to either nsPEF or sham exposure at room temperature (22 ± 2 °C). Trains of 200 ns pulses were delivered to cuvettes from an AVTECH AVOZ-D2-B-ODA generator (AVTECH Electrosystems, Ottawa, Ontario, Canada). Samples were exposed to up to 600, 200 ns, 7 kV/cm pulses at 10 Hz. Sample heating was measured for the highest pulse dose (600, 200 ns, 7 kV/cm, 10 Hz) using a thermocouple thermometer (Physitemp, Clifton, NJ, USA). By the end of PEF exposure, the measured values averaged 0.42 +/− 0.11. 

Once exposure was completed, cells were seeded in 96, 24, or 6 well plates and placed at 37 °C in the incubator for the different incubation times (4 to 24 h). 

### 4.3. Viability and Caspase-3/7 Activity Assays

Viability was measured at 4 and 24 h after nsPEF treatment using the resazurin-based metabolic assay Presto Blue (Life Technologies, Grand Island, NY, USA). Plates were read with a Synergy 2 microplate reader, with excitation/emission settings at 530/590 nm. 

In certain experiments, concurrently with cell viability, caspase-3/7 activity was measured. We first recorded fluorescence (Presto Blue/viability) and, then, added the Caspase-Glo 3/7 assay (Promega Corporation, Madison, WI, USA) according to manufacturer’s instructions. 

Samples were measured in triplicates, the data were averaged, corrected for the background, and considered as a single experiment.

### 4.4. RT-PCR and Western Blot

The RT-PCR procedure was described in detail previously [51]. Gene expression analysis was conducted using TaqMan Gene Expression Assays for spliced *XBP1* (Mm03464496_m1) [104] and the endogenous housekeeping gene *HPRT* (Mm03024075_m1). Relative fold-changes were calculated using the 2^−ΔΔ^*^Ct^* method. Significance was determined by comparing the 2^−Δ^*^Ct^* value using a one-way analysis of variance with a Dunnett’s post hoc test. Error bars represent the standard deviation of the relative-fold expression between samples.

For Western blot, membranes were blocked in TBS 5% BSA buffer for 1 h at room temperature. Primary rabbit anti-phospho-eIF2α (Serine 51) and rabbit anti-eIF2α antibodies and secondary anti-rabbit IgG HRP were from Cell Signaling (Cell Signaling Technology, Danvers, MA USA). Rabbit anti-vinculin was from Abcam (Abcam, Cambridge, UK). Antibodies were diluted in TBS, 5% BSA, 0.2% Tween buffer according to the manufacturer’s instructions. The membranes were incubated overnight at +4 °C and for 1 h at room temperature with primary and secondary antibodies, respectively. Membranes were imaged using a ChemiDoc MP imaging system (BioRad, Hercules, CA, USA). Blot images in Figure 1 have been cropped, full-length blots are presented in Appendix A.

### 4.5. DAMPs Detection

Cell membrane CRT was measured at 24 h post nsPEF by flow cytometry using a rabbit polyclonal anti-CRT antibody (Abcam) and a secondary goat anti-rabbit IgG (H + L) 488 (Invitrogen, Carlsbad, CA, USA). Dead cells staining positive for DAPI were excluded from the analysis. Samples were acquired using a MACSQuant Analyzer 10 flow cytometer (Miltenyi Biotec, Bergisch Gladbach, Germany) and analyzed with FlowJo software (FlowJo, Ashland, OR, USA).

HMGB1 was measured at 24 h post treatment by ELISA according to manufacturer’s instructions (IBL International, Hamburg, Germany). ATP was measured at 18 h post nsPEF using the ENLITEN ATP assay according to manufacturer’s instructions (Promega). Plates were read with a Synergy 2 microplate reader. Samples were measured in triplicates, averaged, and considered as a single experiment.

### 4.6. Vaccination Experiments 

Typically, millions of dead cells are used in vaccination experiments, however these numbers are not feasible when using 1 mm electroporation cuvette to kill tumor cells. In preliminary experiments aimed at optimizing the conditions for our vaccination experiments, we established that 0.6 × 10^6^ doxorubicin treated CT26 cells were enough to vaccinate a mouse by a challenge dose of 0.1 × 10^6^ live cells. For the poorly immunogenic EL-4 model we decided to keep 0.6 × 10^6^ dead cells for the vaccine but to lower the challenge dose to the minimum number of cells needed to establish an EL-4 tumor, which we found to be 0.03 × 10^6^ cells. 

Both CT26 and EL-4 cells were resuspended at 3 × 10^6^/mL in PBS and aliquoted into the electroporation cuvettes (about 3 cuvettes/animal). Immediately after the treatment, all samples were pooled together, cells were counted, centrifuged, resuspended at 6 × 10^6^/mL in PBS and injected in vivo in syngeneic mice (100 µL mouse).

As a positive control for ICD induction, CT26 and EL-4 cells were treated with doxorubicin (25 µM) and mitoxantrone (0.5 and 1 µM), respectively. After 24 h, cells were washed three times with PBS, resuspended similarly to nsPEF treated cells and injected in vivo. 

After one week, animals were challenged into the opposite flank with either 0.1 × 10^6^ CT26 or 0.03 × 10^6^ EL-4 live cells and monitored for the appearance of palpable tumors on both flanks. Tumors were measured twice weekly using a caliper. The formula *v* = ab^2^π/6 was used to calculate tumor volumes where a is the longest diameter and *b* is the diameter perpendicular to a.

Seven- to 8-week-old C57/BL6 and BalbC female mice (Jackson Laboratory, Bar Harbor, ME, USA) were housed in individually ventilated cages in groups of 5 under pathogen-free conditions. All procedures were carried out in accordance with the Guide for the Care and Use of Laboratory Animals, Eighth Edition and were approved by the Old Dominion University Institutional Animal Care and Use Committee (permit number: 18-018).

### 4.7. Statistical Analysis

Data are presented as mean ± standard error for *n* independent experiments. Statistical analyses were performed using a two-tailed *t* test where *p* < 0.05 was considered statistically significant. Statistical calculations were accomplished using Grapher 11 (Golden Software, Golden, CO, USA).

## 5. Conclusions

Altogether, we have demonstrated that nsPEF induce ER stress accompanied by ICD. Future research should focus on tailoring the pulse treatment to boost the immunogenicity of tumor cell death across multiple tumor cell types. 

## Figures and Tables

**Figure 1 cancers-11-02034-f001:**
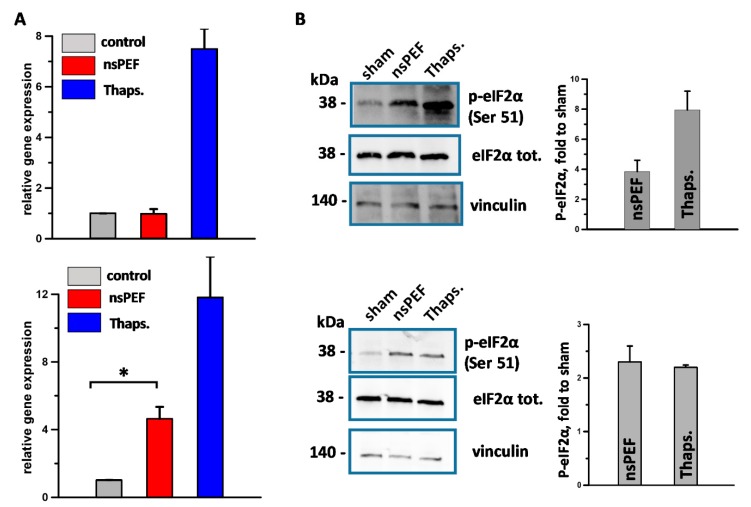
Effect of nsPEF on the activation of the endoplasmic reticulum (ER) stress sensors IRE1 (**A**) and PERK (**B**). EL-4 cells (top panels) and CT26 cell (bottom panels) were treated with iso-effective doses of 100 and 300 pulses, respectively (200 ns, 7 kV/cm, 10 Hz). Samples were collected at 5 h post treatment. In (A) the expression level of *XBP1* in both EL-4 and CT26 was measured by real-time quantitative PCR. The gene mRNA level was normalized to the housekeeping *HPRT* gene mRNA and is shown as relative expression. In (**B**) phosphorylation of eIF2α was measured by Western blot using an anti-phospho-eIF2α (Serine 51) antibody. Left panels show a representative image for both EL-4 (top panel) and CT26 cells (bottom panel) with eIF2α (phosphorylated and total) and the housekeeping Vinculin protein seen as a 38 and 140 kDa band, respectively. Graphs on the right are the quantifications of the p-eIF2α expressed as fold to sham. 1 µM thaspigargin (Thaps.) was used as a positive control for ER stress induction. Mean +/− s.e. *n* = 3 for both A and B. * *p*  <  0.001 for the difference of nsPEF from sham.

**Figure 2 cancers-11-02034-f002:**
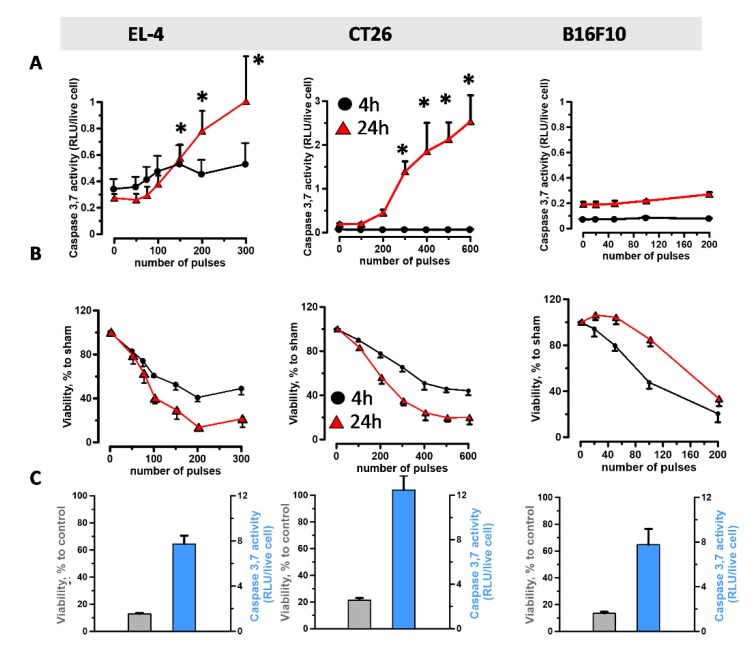
nsPEF trigger apoptotic cell death in both CT-26 and EL-4 cells but not in B16F10 melanoma cells. Cells were exposed in cuvettes to increasing numbers of 200 ns pulses (7 kV/cm, 10 Hz). Both caspase-3/7 activation (**A**) and cell viability (**B**) were measured at 4 h (black) and 24 h (red) post treatment. Caspase activity is shown as relative luminescence units (RLU) per live cell while cell viability is expressed in % to sham-exposed parallel control. In (**C**) cells were treated with 10 µM staurosporin for 24 h. The left y-axis refers to cell viability (grey), and the right y-axis is the caspase activity (blue). Mean +/− s.e. *n*  =  3–5. * *p*  < 0.05, ** *p*  <  0.01 for caspase activity of nsPEF from sham.

**Figure 3 cancers-11-02034-f003:**
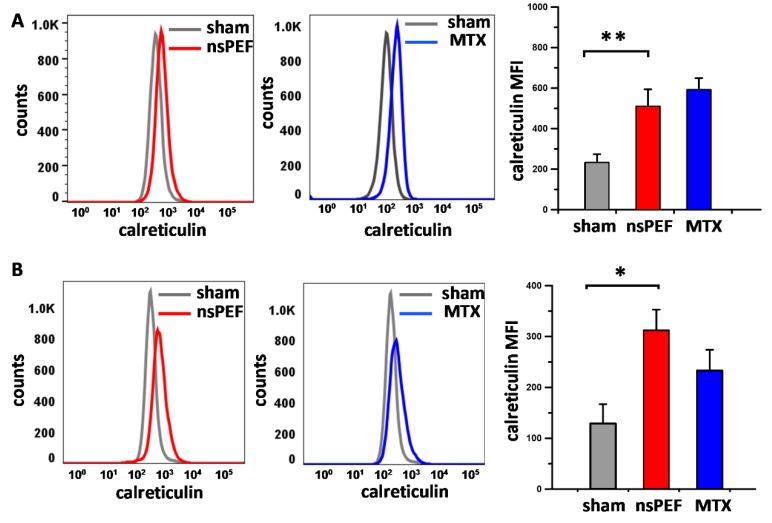
nsPEF induce CRT externalization on the cell surface of both EL-4 (**A**) and CT26 cells (**B**). EL-4 and CT26 cells were treated with iso-effective doses of 100 and 300 pulses, respectively (200 ns, 7 kV/cm, 10 Hz). 1 µM mitoxantrone (MTX) was used as a positive control for CRT exposure. At 24 h post-treatment, surface CRT was measured by FACS analysis. Left and middle panels show representative FACS histograms. Bar graphs on the right are quantifications of the surface CRT expressed as mean fluorescence intensity (MFI). Mean +/− s.e. *n* = 4 for both (**A**) and (**B**). * *p*  <  0.01, ** *p*  <  0.001 for the difference of nsPEF from sham.

**Figure 4 cancers-11-02034-f004:**
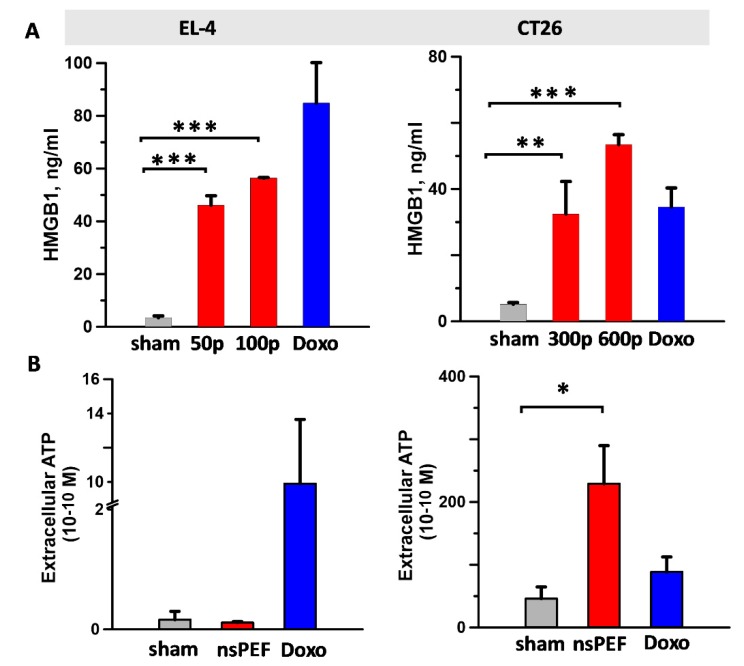
Effect of nsPEF on HMGB1 (**A**) and ATP (**B**) release. EL-4 and CT26 cells were treated with (**A**) the indicated number of pulses or (**B**) iso-effective doses of 100 and 300 pulses, respectively (200 ns, 7 kV/cm, 10 Hz). In (**A**) 24 h post treatment, supernatants were assessed for HMGB1 by ELISA. In (**B**) ATP release in the supernatant was measured at 18 h post treatment using a luciferase-based assay. Doxorubicin (10 µM) was used as a positive control for both HMGB1 and ATP release. Mean +/− s.e. *n* = 4–5 and 3–5 for (**A**) and (**B**), respectively. * *p*  <  0.05, ** *p* < 0.01, *** *p* < 0.001 for the difference of nsPEF from sham.

**Figure 5 cancers-11-02034-f005:**
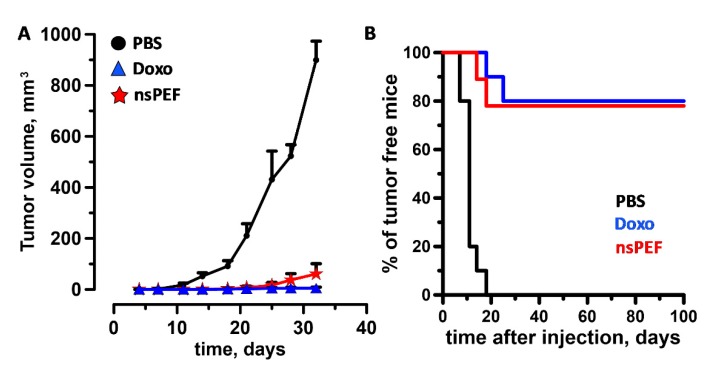
nsPEF-treated CT26 cells vaccinated mice from tumor challenge. CT26 tumor cells were treated with nsPEF (600, 200 ns, 7 kV/cm, 10 Hz) and immediately injected into the flank of syngeneic BalbC mice (0.6 × 10^6^ cells/mouse). Control groups were vaccinated with either PBS or with CT-26 cells treated with doxorubicin (Doxo, 25 µM) for 24 h. After 7 days, animals were challenged with live cells (0.1 × 10^6^ cells/mouse) into the opposite flank. Panel (**A**) shows the tumor growth curves and (**B**) the % of tumor free animals. Graphs show the effects measured in animals that did not develop tumor at the vaccination site. Mean +/− s.e., *n*  =  10, 10, and 9 for PBS, Doxo and nsPEF groups, respectively.

**Figure 6 cancers-11-02034-f006:**
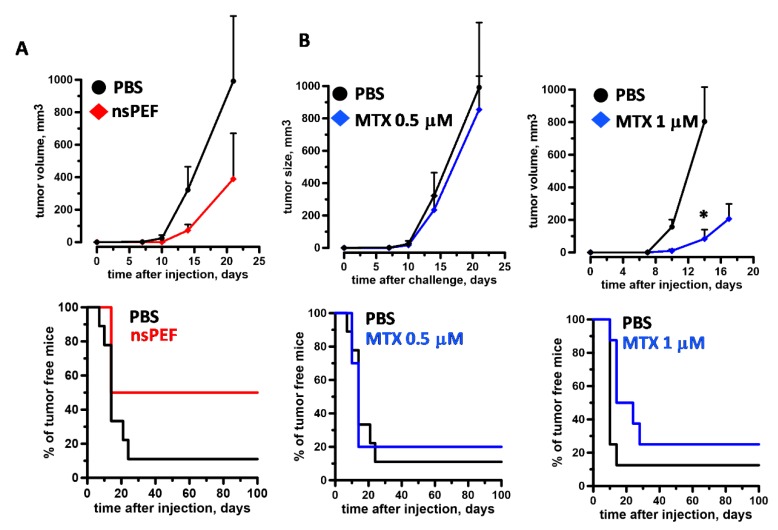
Immunogenicity of killed EL-4 cells: nsPEF vs. mitoxantrone. In (**A**) EL-4 tumor cells were treated with nsPEF (200, 200 ns, 7 kV/cm, 10 Hz) and immediately injected into the flank of syngeneic C57BL6 mice (0.6 × 10^6^ cells/mouse). In (**B**) cells were treated with either 0.5 µM (B, left graphs) or 1 µM (B, right graphs) mitoxantrone (MTX) for 24 h. Control groups were vaccinated with PBS. After 7 days, animals were challenged with live cells (0.03 × 10^6^ cells/mouse) into the opposite flank. Top graphs show the tumor growth curves and bottom graphs show % of tumor free animals. Graphs show the effects measured in animals that did not develop tumor at the vaccination site. Mean +/− s.e., *n*  =  9 and 6 for PBS, and nsPEF groups in (**A**), respectively. Mean +/− s.e., *n* = 9, 9, 8, and 8 for PBS, 0.5 µM MTX, PBS, and 1 µM MTX groups in (**B**), respectively. * *p* < 0.01.

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
