# Peer review of "Nanosecond Pulsed Electric Fields Induce Endoplasmic Reticulum Stress Accompanied by Immunogenic Cell Death in Murine Models of Lymphoma and Colorectal Cancer"

_cancers, 2019, doi:10.3390/cancers11122034_

Round 1
Reviewer 1 Report
This manuscript claims nanosecond pulsed electric fields induce endoplasmic reticulum stress, which has accompanied by immunogenic cell death in murine models of lymphoma and colorectal cancer.
This work has many interesting points to nanosecond electric field induced immunogenic cell death, hence I recommend this manuscript might be published to journal after some of following revisions;
The mechanism for this immunogenic cell death by nsPEF might be OK, however, is there any physical or chemical reasons for this related to reactive oxygen species or nitrogen species? Authors are requested to include this point in the analysis for reader's understandings. The magnitude of nanosecond electric fields in the cell should be estimated for reader's understandings in the text.Author Response
Point-by-point response to Reviewer 1
We thank the reviewer for her/his interest in our work and useful comments which helped us to improve the quality of our manuscript.
The mechanism for this immunogenic cell death by nsPEF might be OK, however, is there any physical or chemical reasons for this related to reactive oxygen species or nitrogen species?
Whether nsPEF can directly generate reactive oxygen species (ROS) in the cell is unknown. However, it has been reported that nsPEF exposure can damage mitochondria causing leakage of ROS and cytochrome C (1). In the discussion (page 9) we clarified this point. Concerning nitrogen species, we are not aware of any study that has investigated the effect of nsPEF on these species.
The magnitude of nanosecond electric fields in the cell should be estimated for reader's understandings in the text.
We thank the reviewer for this comment. Initially, the electric field inside the cell should be equal to the applied electric field (in our study 7 kV/cm). Once the cell plasma membrane is charged, the electric field will compensate the applied field and therefore protect the cell interior. For mammalian cell membrane charging time constant is in the order of 100 ns. This point has been clarified in the introduction (page 2).
References
Beebe, S.J.; Fox, P.M.; Rec, L.J.; Willis, E.L.; Schoenbach, K.H. Nanosecond, high-intensity pulsed electric fields induce apoptosis in human cells. Faseb J 2003, 17, 1493-1495.

Reviewer 2 Report
The manuscript by Rossi et al demonstrates that ns pulsed electroporation can induce to varying degree molecular hallmarks of ER stress, apoptosis and immunogenic cell death in two murine tumor cell lines in vitro. Furthermore, they demonstrate that vaccination with ns electoporated tumor cells can diminish or delay tumor growth in an allograft mouse model.
In general, the study is convincing, well written and should be of general interest to readers of the journal. However, some concerns should be addressed:
While the data presented does extend some previous findings from other groups, much of the work reiterates work in other cell types. The more novel aspect of the current study is the in vivo work that suggested that ns pulsed tumor cells induced an immunogenic response that diminish tumor growth. As such, some histological analysis and/or assessment of immune response in allograft animals would significantly strengthen the work. At the very least, a detailed description of how tumor volume as assessed is warranted. EL-4 tumor cell line is derived from C57BL6 but CT-26 tumor cells were originally derived in BALB/c mice. Was a possible additional effect on tumor progression in the vaccination experiment considered? Fig2. The effect of ns pulsing on caspase activation is at least three-fold more effective in CT-26 compared to EL-4, yet cell viability is nearly identical between the two tumor lines. Please speculate. Fig4. Basal amounts of secreted ATP are much lower in EL-4 cells. Was ATP content between the two lines determined or secretion measured at earlier timepoints? Why difference in time points post treatment (24 vs 18 hrs) for HMGB and ATP release? Immunogenicity experiments- Why was there a difference in the pulse paradigm used in in vitro and allograft experiments? Line 258, remove “always” from the text
Author Response
Point-by-point response to Reviewer 2
We thank the reviewer for her/his interest in our work and useful comments which helped us to improve the quality of our manuscript.
The more novel aspect of the current study is the in vivo work that suggested that ns pulsed tumor cells induced an immunogenic response that diminish tumor growth. As such, some histological analysis and/or assessment of immune response in allograft animals would significantly strengthen the work. At the very least, a detailed description of how tumor volume as assessed is warranted.
The reviewer is right that the assessment of the immune response is of pivotal importance. However, this is a large amount of new data that we are currently collecting. These results will be included in our next paper which continues the current work and will focus on the immune response triggered by nsPEF treatments.
The assessment of the tumor volume has been included in the material and methods section (page 12).
EL-4 tumor cell line is derived from C57/BL6 but CT-26 tumor cells were originally derived in BALB/c mice. Was a possible additional effect on tumor progression in the vaccination experiment considered?
We thank the reviewer for noticing this. Both EL-4 and CT-26 cells were injected in syngeneic mice, C57BL6 and BalbC, respectively. We have clarified this point in both figure legends 5 and 6 and in material and methods (page 12).
The effect of ns pulsing on caspase activation is at least three-fold more effective in CT-26 compared to EL-4, yet cell viability is nearly identical between the two tumor lines. Please speculate.
We previously reported that nsPEF can concurrently trigger both necrotic and apoptotic pathways (1). Necrotic cell death is due to a loss of membrane integrity leading to water uptake, cell swelling, and eventual membrane rupture. Although we don’t know what controls the balance between the two modes of cell death, we should consider that both cell death pathways contribute to nsPEF cyctoxicity. We therefore can speculate that the same level of cell death measured in CT26 and EL-4 treated tumor cells is due to the contribution of primary necrosis.
This point has been included in the result section (page 5).
Basal amounts of secreted ATP are much lower in EL-4 cells. Was ATP content between the two lines determined or secretion measured at earlier timepoints?
Although the reviewer is right that the amount and/or the kinetics for ATP release in the two cell lines might be different one should consider that 1. during ICD ATP is released through active mechanisms (i.e. autophagy) and is usually measured at 18-24 h post treatment. 2. ATP can passively be released through nanopores (2) and therefore early measurements can lead to misleading interpretation. This point has been clarified in the discussion (page 10).
Why difference in time points post treatment (24 vs 18 hrs) for HMGB and ATP release?
The time points were chosen based on published data (3-5). ATP is considered an early apoptotic DAMP and is usually measured at 18-24 h post-treatment while HMGB1 is a late apoptotic DAMP release during secondary necrosis and is measured at 24-48 h post treatment.
Immunogenicity experiments- Why was there a difference in the pulse paradigm used in in vitro and allograft experiments?
DAMPs were detected using iso-effective doses causing 60 % cell death or less. These doses in in vivo studies would have caused tumor growth at vaccination site in 100 % of the animals therefore impairing the study. This point has been clarified in the result section (page 7) and in the discussion (page 10).
Line 258, remove “always” from the text
This modification has been implemented (see line 256).
References
Pakhomova, O.N.; Gregory, B.W.; Semenov, I.; Pakhomov, A.G. Two modes of cell death caused by exposure to nanosecond pulsed electric field. PLoS One 2013, 8, e70278, doi:10.1371/journal.pone.0070278. Rols, M.P.; Teissie, J. Electropermeabilization of mammalian cells. Quantitative analysis of the phenomenon. Biophys J 1990, 58, 1089-1098, doi:10.1016/S0006-3495(90)82451-6. Kepp, O.; Senovilla, L.; Vitale, I.; Vacchelli, E.; Adjemian, S.; Agostinis, P.; Apetoh, L.; Aranda, F.; Barnaba, V.; Bloy, N., et al. Consensus guidelines for the detection of immunogenic cell death. Oncoimmunology 2014, 3, e955691, doi:10.4161/21624011.2014.955691. Garg, A.D.; Dudek, A.M.; Agostinis, P. Cancer immunogenicity, danger signals, and DAMPs: what, when, and how? Biofactors 2013, 39, 355-367, doi:10.1002/biof.1125 Martins, I.; Wang, Y.; Michaud, M.; Ma, Y.; Sukkurwala, A.Q.; Shen, S.; Kepp, O.; Metivier, D.; Galluzzi, L.; Perfettini, J.L., et al. Molecular mechanisms of ATP secretion during immunogenic cell death. Cell Death Differ 2014, 21, 79-91, doi:10.1038/cdd.2013.75.